# Different Causal Factors Occur between Land Use/Cover and Vegetation Classification Systems but Not between Vegetation Classification Levels in the Highly Disturbed Jing-Jin-Ji Region of China

**Sangui Yi [1,2], Jihua Zhou [1], Liming Lai [1], Qinglin Sun [1,2], Xin Liu [1,2], Benben Liu [1,2], Jiaojiao Guo [1,2] and Yuanrun Zheng [1,*]**

1   Key Laboratory of Resource Plants, Institute of Botany, Chinese Academy of Sciences, Beijing 100093, China; m15874232874@163.com (S.Y.); zhoujihua@ibcas.ac.cn (J.Z.); lailiming@ibcas.ac.cn (L.L.); sunqinglin@ibcas.ac.cn (Q.S.); liuxbest@126.com (X.L.); liubenben@ibcas.ac.cn (B.L.); guojj169@163.com (J.G.)

2   Key Laboratory of Resource Plants, Institute of Botany, Chinese Academy of Sciences, University of Chinese Academy of Sciences, Beijing 100049, China

*   Correspondence: zhengyr@ibcas.ac.cn

**Abstract:** Land use/cover and vegetation patterns are influenced by many ecological factors. However, the effect of various factors on different classification systems and different levels of the same system is unclear. We conducted a redundancy analysis with 10 landscape metrics and ecological factors in four periods (1986–2005/2007, 1991–2005/2007, 1996–2005/2007, 2001–2005/2007) to explore their effects on the land use/cover system, vegetation group and vegetation type, and formation and subformation levels of the vegetation classification system in the Jing-Jin-Ji region. Soil, temperature and precipitation from 1986–2005, 1991–2005, and 2001–2005 were the important causal factors, and anthropogenic disturbance and atmospheric factors in 1996–2005 were causal factors at the land use/cover level. The total explained variance from 1996–2005 and 2001–2005 was higher than that from 1986–2005 and 1991–2005 at the land use/cover level. Causal factors and the variance explained by causal factors at the vegetation group, vegetation type, and formation and subformation levels were similar but different in the land use/cover system. Geography, soil and anthropogenic disturbance were the most important causal factors at the three vegetation levels, and the total explained variance from 2001–2007 was higher than that from 1986–2007, 1991–2007, and 1996–2007 at the three vegetation levels. In environmental research, natural resource management and urban or rural planning, geographic factors should be considered at the vegetation group, vegetation type and formation and subformation levels while atmospheric and temperature factors should be considered at the land use/cover level.

**Keywords:** pattern of vegetation and land use/cover; landscape metrics; ecological factors; redundant analysis

## 1. Introduction

Vegetation is an important component of ecosystems that can provide various service functions for human beings. Vegetation classification mainly focuses on the vegetation itself; types of vegetation in different classification levels must be taken into account in environmental research, natural resource management and urban or rural planning [1]. Closely related to vegetation, land use/cover is a complex formed by the interaction of natural and artificial elements on the Earth's surface, including various types of natural covers (such as soil and natural vegetation) and artificial covers (such as buildings and roads). Therefore, vegetation classification is much different from land use/cover system, and one type in land use/cover includes several types in the vegetation classification, such as woodland in the classification of land use/cover, which includes needleleaf forest, broadleaf forest, and

scrub in the vegetation classification [2]. Vegetation and land use/cover analysis has been an important topic in ecological research for a long time [1,3,4]. The impact of vegetation and land use/cover change on climate, ecosystem processes, biogeochemical cycles and biodiversity is a major driver of global environmental change [3,5,6]. Both natural and human-induced changes in vegetation and land use/cover are in turn affected by climate, ecosystem processes, biogeochemical cycles and biodiversity [7]. Therefore, determining the relative contributions of factors to the patterns and dynamics of vegetation and land use/cover in some classifications is the first step for the development of sustainable environmental management [8]. Understanding the relationship between vegetation, land use/cover in these classifications, and ecological factors can quantitatively reveal their interactions, explore the main factors leading to vegetation and land use/cover change, and predict the impact of climate change on them [9,10], which provide important information for scientific resource management and decision-making for better human activities [4].

Most studies on vegetation and its causal factors were performed using long-term series parameter data, which can reflect the status of vegetation, such as the Normalized Difference Vegetation Index (NDVI), Leaf Area Index (LAI), Aboveground Biomass (AGB) and Enhanced Vegetation Index (EVI) [11–13]. Studies of land use/cover change and its causal factors are usually based on specific land use/cover type conversion, conversion areas [3,14], and landscape metrics [4]. These studies generally directly analyze the correlation between the changes in vegetation or land use/cover over a long time and their driving factors. Long-term changes in vegetation are driven by multiple interacting biogeochemical drivers and land use effects, including climate [15,16], topography [4], soil [17], and disturbance [18]. Land use/cover is also influenced by these ecological factors [7] because the change in land use/cover is based on human interference and the possibility of its type and mode conversion controlled by the regional natural geographical environment. Studies involving the relationships between these factors and vegetation or land use/cover have been based on representative factors, such as annual precipitation, mean annual temperature, and NDVI [10,19]. However, these representative factors may not be sufficient to accurately simulate the vegetation distribution, and many more factors may be required under reasonable situations [2].

Methods used to reveal the relationship generally include ordination techniques and related statistical models [17,20,21]. Ordination analysis, as an effective method, is usually used to explore and quantify the impact of explanatory variables on response variables, and the following basic ordination methods are often used, e.g., constrained ordinations (redundancy analysis (RDA) and canonical correspondence analysis (CCA)), unconstrained ordinations (principal component analysis (PCA), correspondence analysis (CA), principal coordinate analysis (PCoA), and non-metric multi-dimensional scaling (NMDS)), eigenvalue-based ordinations (such as PCA, PCoA, CA, RDA, and CCA), distance-based ordinations (such as NMDS), linear response models (such as PCA and RDA) and unimodal models (such as CCA and CA) [22,23]. These statistical tools are used at a much smaller level—the patch level, and examine relationships between species composition and local environmental factors around the world, such as in the Azores archipelago, Europe, and Asia [17,24–26], and the results show that vegetation is influenced by many environmental factors, including soil, lithology, elevation, latitude, seasonality, temperature, precipitation, wind, cloud, light, and human historical disturbance [9,20,25–27]. Related statistical models, such as linear models [13,28], correlation analysis [29], regression analysis [11,15], attribution analysis [14], and other mathematical models [21], are usually used to explore and quantify the impact of specific factors on an object in the context of more specific objectives around the world, such as Asia, Europe, and Oceania, and they are not only used to research the relationship between the environment and vegetation [14,21,29] but also to investigate the environment–organism relationships and organism–organism relationships [28,30].

Ecological factors are not only related to the composition of special patch types but also to their spatial combination, i.e., landscape pattern [31]. Landscape metrics can

highly concentrate landscape pattern information and reflect some structural composition and spatial configuration characteristics. These metrics at different levels often serve as indicators to quantify vegetation distributions [28,31,32]. For a long time, they have been used in similar studies to objectively describe the patterns of vegetation and land use/cover [4,31]. They are usually used in combination with ordination analysis and related statistical models to investigate these relationships, which is essential for the next or future steps [19,28]. However, most of the associated studies are only performed at one level, such as land use/land cover [14,19] and certain vegetation types [4,32], using one or a few metrics. Because a few metrics can only reflect the vegetation pattern in limited aspects, many more metrics that can cover the overall picture of vegetation at the different levels are necessary. Furthermore, with economic development, understanding the vegetation pattern and its causal factors in highly disturbed regions is of cardinal significance.

The Beijing–Tianjin–Hebei region, also called the Jing-Jin-Ji region, is an important political, cultural, and economic region in China. However, due to rapid economic development since the reform and opening up of China, the environment in the Jing-Jin-Ji region has been heavily disturbed. In addition, the development of this region is extremely uneven. Large cities, including Beijing and Tianjin, are very developed, while other cities are not. Beijing presents the most developed economy, the best social security and the most complete infrastructure; however, it also has the most crowded population, the least per capita possession of natural resources and the worst environmental bearing capacity [33]. To achieve sustainable development in the Jing-Jin-Ji region, the Chinese government led major afforestation projects over the last years [34]. It is necessary to break up the administrative divisions and understand the relationships between vegetation as well as land use/cover and their causal factors from the perspective of the entire region to provide support and insights for further planning and management of natural resources, such as afforestation [34,35]. Moreover, as a world-level metropolitan area, both vegetation and land use/cover have been heavily affected by human disturbance in the Jing-Jin-Ji region; thus, understanding the causal factors for vegetation and land use/cover will provide a reference to balance socioeconomic development and environmental sustainability worldwide.

In this study, we performed a redundancy analysis to determine the causal factors in four time intervals for the pattern of land use/cover, which is often used in land resource management, and vegetation patterns in different class levels, which are often used in vegetation and biodiversity management. We hypothesized that short-term factors have greater impacts on land use/cover pattern than on vegetation pattern. The objective was to (1) investigate the correlations between ecological factors in four time intervals and land use/cover and vegetation metrics on three levels; (2) determine the main causal factors for the change in vegetation and land use/cover; and (3) determine whether short- or long-term average ecological factors have a greater effect on land use/cover and vegetation patterns. The findings would be useful for achieving sustainable development by eliminating the effect of human disturbance on the environment.

## 2. Data and Method

### 2.1. Study Area

The Jing-Jin-Ji region extends from 113°04′ to 119°53′ E and 36°01′ to 42°37′ N and is located in the northern area of the North China Plain, with the Yanshan Mountains to the north, the North China Plain to the south, the Taihang Mountains to the west and Bohai Bay to the east. The Yanshan–Taihang mountain system in the northwest transforms gradually into a plain in the southeast, which shows the characteristics of higher elevation in the northwest and lower elevation in the southeast (Figure 1). It has a temperate monsoon climate, with annual precipitation ranging from 305 to 711 mm and mean annual temperature ranging from −3 to 14 °C (climate data were obtained from WorldClim, http://www.worldclim.org/, accessed on 13 October 2019). The precipitation gradually increases from northwest to southeast, while the temperature gradually declines from northwest to southeast. The southeast is a coastal alluvial plain dominated by agricultural

and urban ecosystems, while forests are mainly distributed in the Yanshan and Taihang Mountains in the northwest. The region covers an area of approximately 216,000 km² and has a population of approximately 110 million, and it includes two provincial cities, Beijing and Tianjin, and 11 prefecture-level cities in Hebei Province [35].

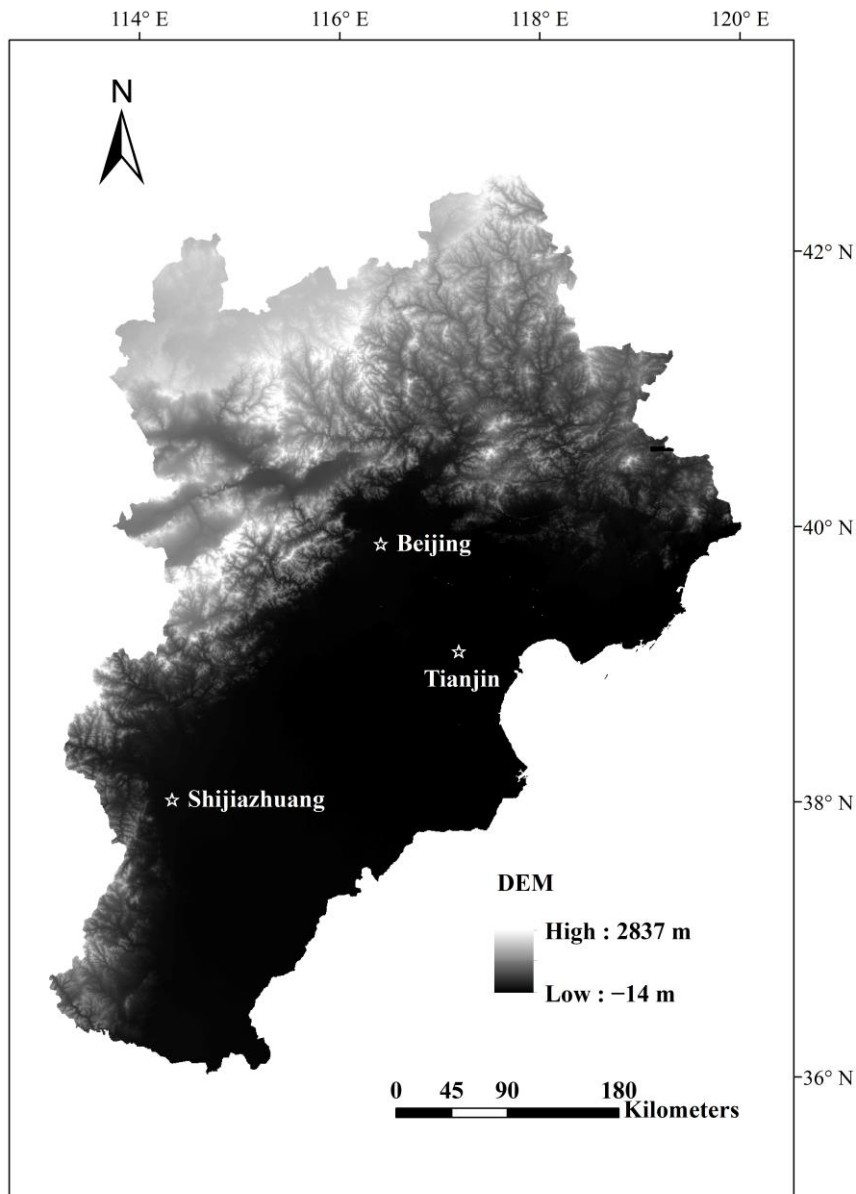

**Figure 1.** The location and Digital Elevation Model (DEM) of the Jing-Jin-Ji region.

### 2.2. Land Use/Cover and Vegetation Category Data and Landscape Metrics

Land use/cover data for the study area in 2005 were obtained from the Resource and Environment Data Cloud Platform (http://www.resdc.cn/Default.aspx, accessed on 13 October 2019), and the vegetation distribution data in 2007 were obtained from the Vegetation Map of the People's Republic of China [1]. Three classification levels of the vegetation distribution data from coarse to fine are vegetation groups (such as needleleaf forest), vegetation types (such as cold-temperate and temperate mountains needleleaf forest), and formations and subformations (such as Larix principis-rupprechtii forest). This region contained six land use/land covers, nine vegetation groups, 16 vegetation types and 58 formations and subformations (Table A1). Considering the suitable spatial scale and amount of sample based on the actual situation, land use/cover and every vegetation level were



split according to 15 scales in ArcGIS 10.3 [2,19] as follows: 10 × 10 km, 12.5 × 12.5 km, 15 × 15 km, 17.5 × 17.5 km, 20 × 20 km, 22.5 × 22.5 km, 25 × 25 km, 27.5 × 27.5 km, 30 × 30 km, 32.5 × 32.5 km, 35 × 35 km, 37.5 × 37.5 km, 40 × 40 km, 42.5 × 42.5 km, and 45 × 45 km. Ten landscape metrics were calculated at each sample at all scales for land use/cover and every vegetation level using Fragstats 4.2 [36]. These metrics included four aggregation characteristics (connectance index, patch density, landscape shape index, and mean Euclidean nearest neighbor distance), three area-edge characteristics (largest patch index, mean patch area, and edge density), two diversity characteristics (patch richness and Shannon's diversity index), and one shape characteristic (mean shape index) [31].

### 2.3. Geographic, Soil, Precipitation, Temperature, Atmospheric, and Anthropogenic Disturbance Data

The ecological factors used in this study involved geographic, soil, precipitation, temperature, atmospheric, and anthropogenic disturbance factors (Table A2). Geographic variables were obtained from Zhao et al. (2018) [37]. The soil data were provided by the China soil map-based harmonized world soil database (HWSD). All attribute data, including aspect, texture class, available water storage capacity, drainage class and soil depth, were changed into numeric data, e.g., 1, 2, 3, etc.

Population, gross domestic product, gross domestic product from primary industry, gross domestic product from secondary industry, and gross domestic product from tertiary industry from 1986–2007 were obtained from Nian.Jian.XiaZe.com (http://nianjian.xiaze.com/info/bjtjnj.html, accessed on 15 October 2019). The variables of gross domestic product per capita and gross domestic product per unit area from 1986–2007 were calculated from the population and gross domestic product data. The Defense Meteorological Satellite Program/Operational Linescan System night light data from 1992–2007 were obtained from the National Centers for Environmental Information (https://ngdc.noaa.gov/eog/download.html, accessed on 15 October 2019). The monthly average temperature and atmospheric variables from 1986–2007 were obtained from the National Meteorological Science Data Center (http://data.cma.cn/, accessed on 15 October 2019). The mean temperature of the warmest and coldest month in 1986–2007 were calculated from the monthly average temperature. The annual precipitation and mean annual temperature in 1986–2007 were provided by the Resource and Environment Data Cloud Platform (http://www.resdc.cn/Default.aspx, accessed on 15 October 2019). The maximum temperature, minimum temperature, and precipitation for every month in 1986–2007 were provided by WorldClim (https://www.worldclim.org/, accessed on 13 October 2019). Other variables associated with precipitation and temperature in 1986–2007 were calculated from the maximum temperature, minimum temperature, and precipitation in every month.

### 2.4. Suitable Spatial Scale and Redundancy Analysis in Four Periods

We calculated the mean values of ecological variables, including precipitation, temperature, and atmospheric and anthropogenic disturbance factors, at different times. To reveal variables in how a long period of time can affect the pattern of vegetation and land use/cover, the variables in four time intervals were used for land use/cover analysis, i.e., 1986–2005 (except the Defense Meteorological Satellite Program/Operational Linescan System night light), 1991–2005, 1996–2005, and 2001–2005; for vegetation analysis, the four time intervals were 1986–2007 (except the Defense Meteorological Satellite Program/Operational Linescan System night light), 1991–2007, 1996–2007, and 2001–2007 (Table A2).

All variables were natural logarithmically transformed for a normal distribution [2,19]. Landscape metrics for land use/cover and vegetation classification systems served as response variables, and ecological factors served as explanatory variables. Detrended correspondence analysis (DCA) was performed for landscape metrics to detect the length of the species gradient. Because the length of the species gradient was less than 3, a redundancy analysis (RDA) was performed using CANOCO 4.5 to detect the correlation between species variables and environmental variables. As a common and effective method,

RDA is a constrained ordination technique that combines regression analysis and principal component analysis, which can find the best explanatory variables showing the changes of species variables based on the existing environmental variables [2,23]. The suitable spatial scale in land use/cover and each vegetation level were selected according to the explained variability and the sample number at 15 spatial scales [2,19]. Ecological factors with variance inflation factors greater than 10 were removed to avoid correlations among all ecological variables [38]. The significance of the canonical axes was assessed by the Monte Carlo permutation test, and the relative importance of environmental variables and variance explained by them were determined by forward selection [23].

## 3. Results

### 3.1. Suitable Spatial Scale

The number of samples ranged from 107 to 2150 at all scales in the land use/cover system and three levels of the vegetation classification system. Landscape metrics were calculated at a series area extent to determine the optional spatial scale in the four classifications. The 15 × 15 km distance was found to be the suitable optional extent, with more than 40% of the explained variability and of the total sample number in land use/cover. The 12.5 × 12.5 km distance was found to be the suitable optional extent in three vegetation classifications with more than 60% of the explained variability and of the total sample number in three vegetation classifications (Figures A1–A4).

### 3.2. Causal Factors for Land Use/Cover Patterns and Correlations between Landscape Metrics and Environmental Factors

The sums of all canonical eigenvalues were 0.17, 0.18, 0.26, and 0.27 in 1986–2005, 1991–2005, 1996–2005, and 2001–2005, respectively. The causal factors were the same in 1986–2005 and 1991–2005, i.e., drainage class, annual temperature range, soil depth, and annual precipitation. Drainage class, annual precipitation, wind speed in January, and gross domestic product per capita became causal factors in 1996–2005. Compared with the period of 1986–2005, drainage class, annual temperature range, and soil depth were the same causal factors in 2001–2005, and monthly precipitation in April and monthly precipitation in October became causal factors. Compared with the earlier periods of 1986–2005 and 1991–2005, the total variances explained by the important causal factors in later periods of 1996–2005 and 2001–2005 were higher (Table 1).

**Table 1.** Results of redundancy analysis with forward selection in land use/cover level. Variables with explanatory variance >1% and $p < 0.05$ are shown.

| Periods | Variables | Explained Variance (%) | $p$ Value | Pseudo-$F$ Value |
|---|---|---|---|---|
| 1986–2005 | Drainage class | 8.2 | 0.002 | 84.779 |
| | Temperature annual range | 1.7 | 0.002 | 18.350 |
| | Soil depth | 1.6 | 0.002 | 16.922 |
| | Annual precipitation | 1.2 | 0.002 | 12.822 |
| 1991–2005 | Drainage class | 8.2 | 0.002 | 84.714 |
| | Temperature annual range | 1.8 | 0.002 | 19.156 |
| | Soil depth | 1.6 | 0.002 | 17.028 |
| | Annual precipitation | 1.1 | 0.002 | 11.722 |
| 1996–2005 | Wind speed in January | 15.9 | 0.002 | 179.597 |
| | Annual precipitation | 2.9 | 0.002 | 34.407 |
| | Gross domestic product per capita | 1.1 | 0.002 | 13.374 |
| | Drainage class | 1.1 | 0.002 | 13.045 |
| 2001–2005 | Drainage class | 8.6 | 0.002 | 89.703 |
| | Monthly precipitation in April | 3.1 | 0.002 | 33.206 |
| | Monthly precipitation in October | 7.6 | 0.002 | 89.084 |
| | Temperature annual range | 1.4 | 0.002 | 16.287 |
| | Soil depth | 1.2 | 0.002 | 14.336 |

The largest patch index, Shannon's diversity index, and patch richness were highly correlated with drainage class. The connectance index and mean Euclidean nearest neighbor distance were highly negatively correlated with soil depth. Patch density, connection index, and mean patch area were highly correlated with annual precipitation in 1986–2005, the correlations were basically the same in 1991–2005, and different correlations were detected in 1996–2005 and 2001–2005 (Figure 2).

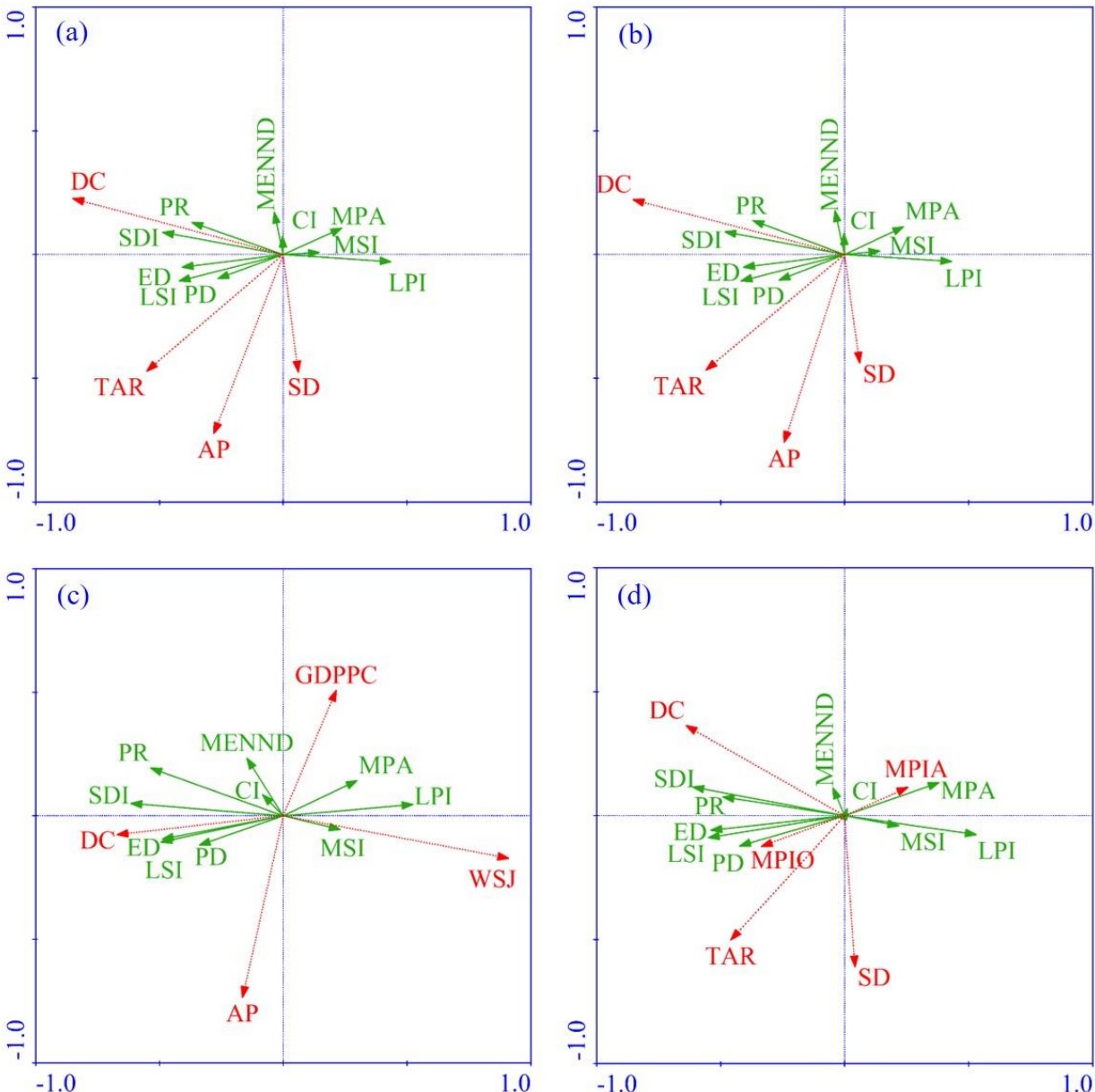

**Figure 2.** Redundancy analysis diagram in the Jing-Jin-Ji region with respect to landscape metrics and environmental factors of land use/cover in 1986–2005 (**a**), 1991–2005 (**b**), 1996–2005 (**c**), and 2001–2005 (**d**). Solid line arrows in green represent landscape metrics, including Largest Patch Index (LPI), Mean Patch Area (MPA), Mean Shape Index (MSI), Connectance Index (CI), Patch Richness (PR), Shannon's Diversity Index (SDI), Patch Density (PD), Edge Density (ED), Landscape Shape Index (LSI), and Mean Euclidean Nearest Neighbor Distance (MENND). Dotted line arrows in red represent environmental factors, including drainage class (DC), soil depth (SD), annual precipitation (AP), temperature annual range (TAR), wind speed in January (WSJ), gross domestic product per capita (GDPPC), monthly precipitation in April (MPIA), and monthly precipitation in October (MPIO).

### 3.3. Causal Factors for Vegetation Group Patterns and Correlations between Landscape Metrics and Environmental Factors

The sums of all canonical eigenvalues were 0.49, 0.50, 0.50, and 0.56 for 1986–2007, 1991–2007, 1996–2007, and 2001–2007, respectively. Slope, drainage class, and soil depth were the top three causal factors at all four time intervals. Gross domestic product per capita was the causal factor only in 1986–2007; Defense Meteorological Satellite Program/Operational Linescan System night light represented the same causal factor in 1991–2007, 1996–2007, and 2001–2007; and monthly precipitation in October, monthly precipitation in April, and gross domestic product from primary industry were only factors in 2001–2007. The total variance explained by causal factors in 2001–2007 was highest at all four time intervals (Table 2).

**Table 2.** Results of redundancy analysis with forward selection in vegetation group level. Variables with explanatory variance >1% and $p < 0.05$ are shown.

| Periods | Variables | Explained Variance (%) | $p$ Value | Pseudo-$F$ Value |
|---|---|---|---|---|
| 1986–2007 | Slope | 30.8 | 0.002 | 616.405 |
| | Drainage class | 9.3 | 0.002 | 215.027 |
| | Soil depth | 4.1 | 0.002 | 102.267 |
| | Gross domestic product per capita | 1.3 | 0.002 | 33.97 |
| 1991–2007 | Slope | 30.9 | 0.002 | 617.506 |
| | Drainage class | 9.3 | 0.002 | 215.355 |
| | Soil depth | 4.1 | 0.002 | 102.471 |
| | Defense meteorological satellite program/operational linescan System night light | 2.4 | 0.002 | 61.289 |
| 1996–2007 | Slope | 30.9 | 0.002 | 617.651 |
| | Drainage class | 9.3 | 0.002 | 215.401 |
| | Soil depth | 4.1 | 0.002 | 102.487 |
| | Defense meteorological satellite program/operational linescan system night light | 2.3 | 0.002 | 59.772 |
| 2001–2007 | Slope | 31.1 | 0.002 | 623.642 |
| | Drainage class | 9.4 | 0.002 | 218.036 |
| | Soil depth | 4.2 | 0.002 | 104.107 |
| | Defense meteorological satellite program/operational linescan system night light | 2.2 | 0.002 | 55.998 |
| | Monthly precipitation in October | 1.2 | 0.002 | 31.666 |
| | Monthly precipitation in April | 3.6 | 0.002 | 102.463 |
| | Gross domestic product from primary industry | 2.0 | 0.006 | 58.153 |

The largest patch index and connectivity index were highly correlated with slope and soil depth in the four periods. The correlations of drainage class to landscape metrics were similar in the four periods (Figure 3).

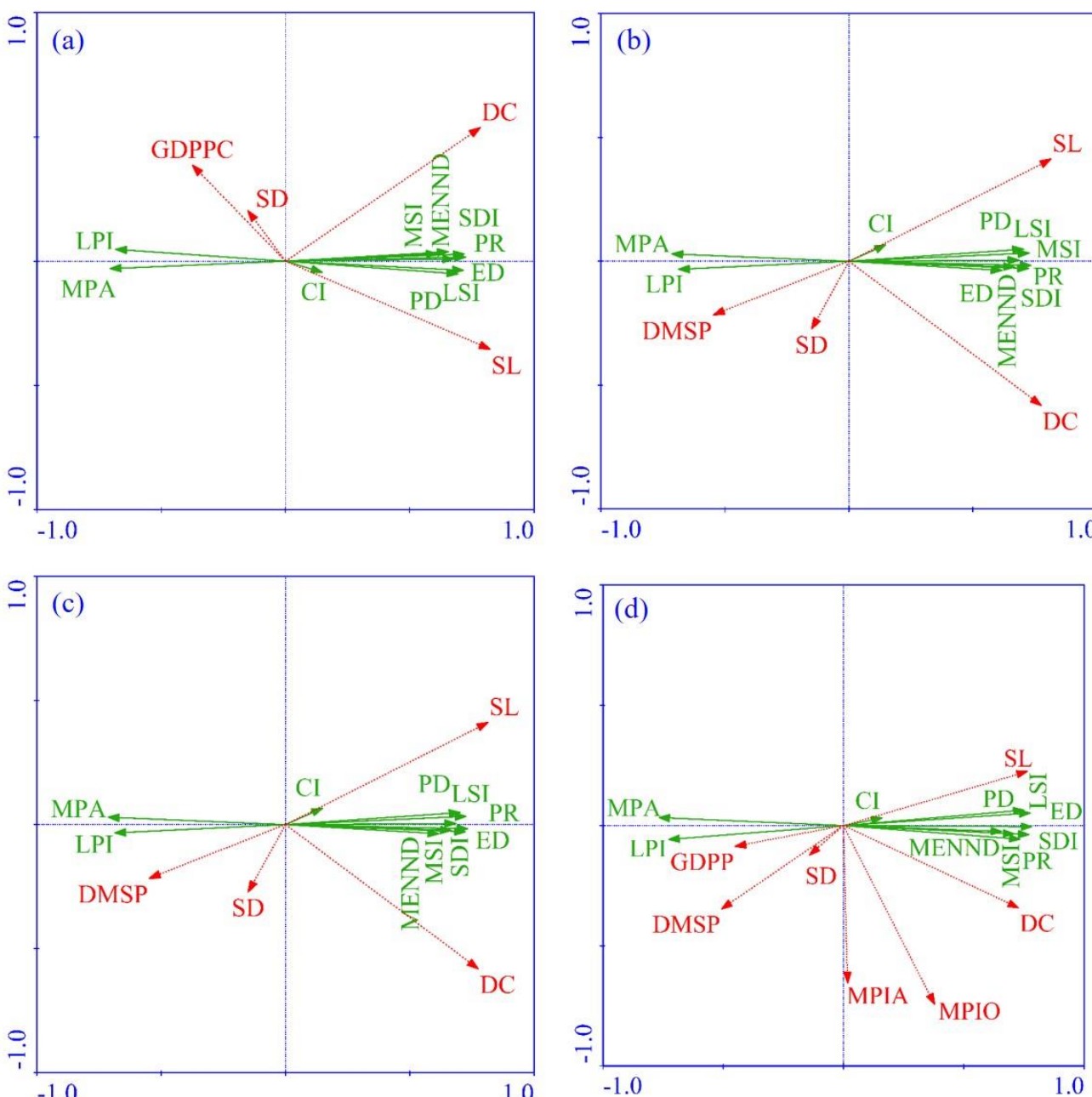

**Figure 3.** Redundancy analysis diagram in the Jing-Jin-Ji region with respect to landscape metrics and environmental factors of vegetation group in 1986–2007 (**a**), 1991–2007 (**b**), 1996–2007 (**c**), and 2001–2007 (**d**). Abbreviations: slope (SL), defense meteorological satellite program/operational linescan system night light (DMSP). Other abbreviations are listed in Figure 2.

### 3.4. Causal Factors for Vegetation Type Patterns and Correlations between Landscape Metrics and Environmental Factors

Similar to vegetation group, the sums of all canonical eigenvalues were 0.49, 0.50, 0.50, and 0.56 in 1986–2007, 1991–2007, 1996–2007, and 2001–2007, respectively. The causal factors in each of the four time intervals were the same at the vegetation group level, and only the explained variance of each factor was different. The total explained variance in 2001–2007 was also higher than that in 1986–2007, 1991–2007, and 1996–2007 (Table 3).

**Table 3.** Results of redundancy analysis with forward selection in vegetation type level. Variables with explanatory variance >1% and $p < 0.05$ are shown.

| Periods | Variables | Explained Variance (%) | $p$ Value | Pseudo-$F$ Value |
|---|---|---|---|---|
| 1986–2007 | Slope | 30.1 | 0.002 | 595.502 |
| | Drainage class | 9.9 | 0.002 | 226.794 |
| | Soil depth | 4.1 | 0.002 | 100.658 |
| | Gross domestic product per capita | 1.1 | 0.002 | 28.643 |
| 1991–2007 | Slope | 30.1 | 0.002 | 596.233 |
| | Drainage class | 9.9 | 0.002 | 226.991 |
| | Soil depth | 4.1 | 0.002 | 100.784 |
| | Defense meteorological satellite program/operational linescan system night light | 1.9 | 0.002 | 47.904 |
| 1996–2007 | Slope | 30.1 | 0.002 | 596.357 |
| | Drainage class | 9.9 | 0.002 | 227.048 |
| | Soil depth | 4.1 | 0.002 | 100.797 |
| | Defense meteorological satellite program/operational linescan system night light | 1.8 | 0.002 | 46.595 |
| 2001–2007 | Slope | 30.4 | 0.002 | 602.728 |
| | Drainage class | 9.9 | 0.002 | 229.933 |
| | Soil depth | 4.1 | 0.002 | 102.372 |
| | Defense meteorological satellite program/operational linescan system night light | 1.7 | 0.002 | 42.761 |
| | Monthly precipitation in October | 1.1 | 0.002 | 29.045 |
| | Monthly precipitation in April | 3.9 | 0.002 | 111.083 |
| | Gross domestic product from primary industry | 2.0 | 0.002 | 57.988 |

Generally, the correlations between landscape metrics and environmental factors were similar in 1986–2007, 1991–2007, and 1996–2007; the largest patch index and connectivity index were highly correlated with slope at the four time intervals; and the correlations of gross domestic product from primary industry for 2001–2007 were similar to those for 1986–2007 (Figure 4).

*3.5. Causal Factors for Formation and Subformation Patterns and Correlations between Landscape Metrics and Environmental Factors*

Similar to vegetation group and vegetation type, the sums of all canonical eigenvalues were 0.50, 0.51, 0.51, and 0.56 in 1986–2007, 1991–2007, 1996–2007, and 2001–2007, respectively, and the total explained variance in 2001–2007 was higher than that in 1986–2007, 1991–2007, and 1996–2007. The causal factors at each of the four time intervals were the same at the vegetation type level, and only the explained variance of each factor was different (Table 4).

The correlation between gross domestic product from primary industry and drainage class in 1986–2007 was similar to that in 1986–2007 at the vegetation type level; most correlations between landscape metrics and ecological factors in 1991–2007, 1996–2007, and 2001–2007 were basically the same as those in 1991–2007, 1996–2007, and 2001–2007 at the vegetation type level (Figure 5).

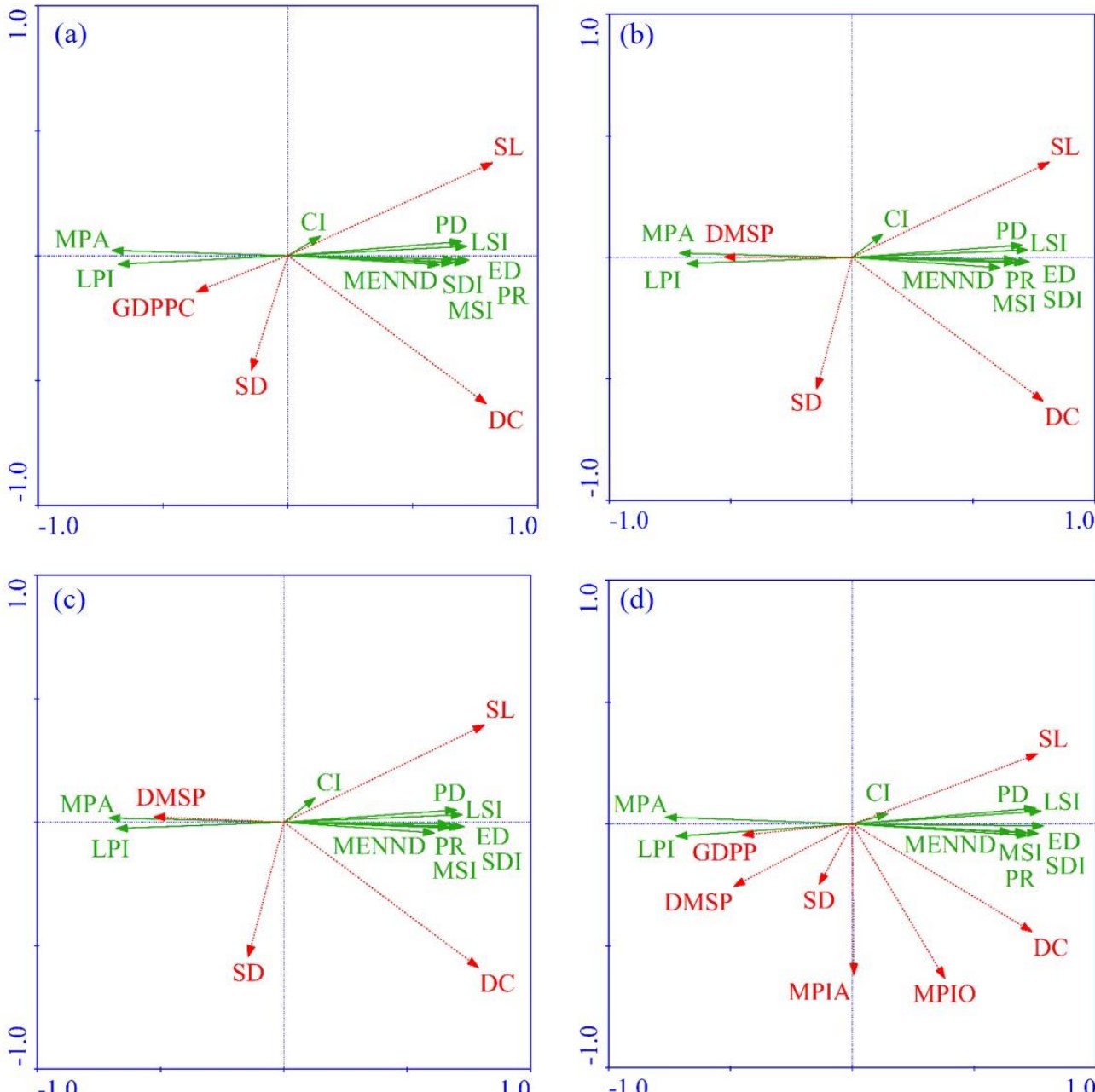

**Figure 4.** Redundancy analysis diagram in the Jing-Jin-Ji region with respect to landscape metrics and environmental factors of vegetation type in 1986–2007 (**a**), 1991–2007 (**b**), 1996–2007 (**c**), and 2001–2007 (**d**). Abbreviations: gross domestic product from primary industry (GDPP). Other abbreviations are detailed in Figures 2 and 3.

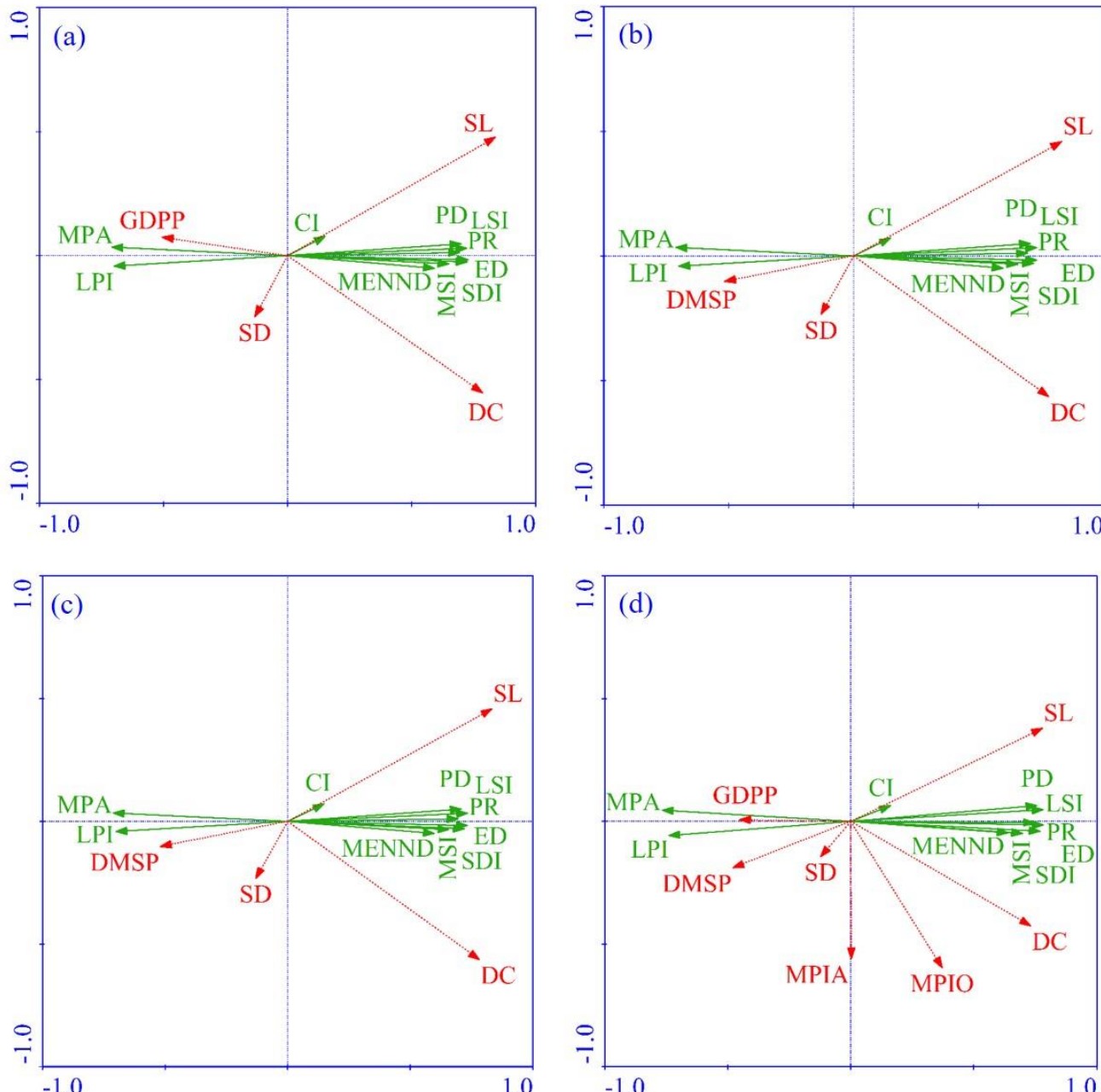

**Figure 5.** Redundancy analysis diagram in the Jing-Jin-Ji region with respect to landscape metrics and environmental factors of formation and subformation in 1986–2007 (**a**), 1991–2007 (**b**), 1996–2007 (**c**), and 2001–2007 (**d**). Abbreviations are detailed in Figures 2–4.

**Table 4.** Results of redundancy analysis with forward selection in formation and subformation level. Variables with explanatory variance >1% and $p < 0.05$ are shown.

| Periods | Variables | Explained Variance (%) | $p$ Value | Pseudo-$F$ Value |
|---|---|---|---|---|
| 1986–2007 | Slope | 32.3 | 0.002 | 659.428 |
| | Drainage class | 9.3 | 0.002 | 218.573 |
| | Soil depth | 3.6 | 0.002 | 90.073 |
| | Gross domestic product from primary industry | 1.1 | 0.002 | 27.775 |
| 1991–2007 | Slope | 32.3 | 0.002 | 660.657 |
| | Drainage class | 9.3 | 0.002 | 218.791 |
| | Soil depth | 3.6 | 0.002 | 90.23 |
| | Defense meteorological satellite program/operational linescan system night light | 1.7 | 0.002 | 43.54 |
| 1996–2007 | Slope | 32.3 | 0.002 | 660.824 |
| | Drainage class | 9.3 | 0.002 | 218.851 |
| | Soil depth | 3.6 | 0.002 | 90.239 |
| | Defense meteorological satellite program/operational linescan system night light | 1.6 | 0.002 | 42.515 |
| 2001–2007 | Slope | 32.6 | 0.002 | 667.178 |
| | Drainage class | 9.3 | 0.002 | 221.472 |
| | Soil depth | 3.6 | 0.002 | 91.715 |
| | Defense meteorological satellite program/operational linescan system night light | 1.5 | 0.002 | 39.188 |
| | Monthly precipitation in October | 1.0 | 0.002 | 26.634 |
| | Monthly precipitation in April | 3.9 | 0.002 | 111.543 |
| | Gross domestic product from primary industry | 2.0 | 0.002 | 59.899 |

## 4. Discussion

### 4.1. Factors Affecting Landscape Patterns in Different Classification

The combination of various ecological factors contributes to the distribution of natural vegetation communities and then changes the landscape [24]. Geography usually serves as an important impact factor in vegetation and is used to predict the vegetation distribution [10,39,40]. In this study, geography was an important impact factor for the vegetation pattern at the three vegetation classification levels during four periods (1986–2007, 1991–2007, 1996–2007, 2001–2007), and geography could much better explain the pattern at the three vegetation classifications than land use/cover during four periods (1986–2005, 1991–2005, 1996–2005, 2001–2005) (Tables 1–4). Many researchers have shown that the spatial heterogeneity of soil characteristics, such as soil moisture, soil organic carbon, total nitrogen, electrical conductivity, pH, soil depth, soil drainage, and C:N ratios, is highly related to vegetation [8,29,41,42]. In this study, drainage class and soil depth were found to have a significant impact on vegetation patterns at the three vegetation classification levels at four time intervals (1986–2007, 1991–2007, 1996–2007, 2001–2007). The drainage class of soil also had a significant impact on landscape pattern at the land use/cover level during four periods (1986–2005, 1991–2005, 1996–2005, 2001–2005), and the soil depth only had a significant impact on landscape pattern at the land use/cover level during three periods (1986–2005, 1991–2005, 2001–2005) (Tables 1–4). Our results are partly consistent with the studies of Franz et al. (2011) [41] and Motzkin et al. (1999) [8]. The slope, drainage class, and soil depth, as the top three impact factors in vegetation classification, had a relatively stable influence on vegetation patterns at different times, which may be related to the large area of farmland in the Jing-Jin-Ji region. Although many studies have highlighted the importance of human disturbance, such as land use, the development of

agriculture and animal husbandry, to vegetation [8,24,42,43], some studies have shown that within an agro-pastoral ecotone in semiarid regions, the impact of human activities on landscape metrics is weak [19]. In this study, anthropogenic disturbance had a significant effect on the three vegetation levels during four periods (1986–2007, 1991–2007, 1996–2007, 2001–2007). Anthropogenic disturbance had a significant effect on the landscape pattern of land use/cover only in 1996–2007 (Table 1–4). Many studies have shown that climate factors, including temperature, precipitation, and atmospheric factors, are highly related to vegetation [9,19,24]. It is emphasized that temperature and precipitation, especially in spring, are the trigger factors for vegetation growth and phenology [16]. Temperature was closely related to the landscape pattern of land use/cover except for 1996–2005 in this research. Contrary to geography, temperature can much better explain the pattern at the land use/cover level than at the three vegetation classification levels (Tables 1–4). This may be related to the effect of urban heat islands with rapid urban development. Precipitation had a significant effect on land use/cover at the four time intervals in this study and a significant effect on the three vegetation levels in 2001–2007 (Tables 1–4). Some studies have also shown the importance of clouds, light, and wind [9,24]. However, atmospheric factors did not significantly affect the vegetation patterns of the three vegetation levels in this study, and wind only significantly affected land use/cover in 1996–2005 (Tables 1–4).

The characteristics of the same vegetation type living with several identical ecological factors differed in different spaces and different periods [44]. In this study, the causal factors, especially environmental factors, were the same for different vegetation classifications and different periods but different for land use/cover in different periods, and the causal factors were different for land use/cover and vegetation classifications. Classification levels should be considered based on the needs of researchers and policy makers when environmental research, natural resource management, and urban or rural plan making are conducted.

Vegetation, especially vegetation net primary productivity in a specific year, can be affected by ecological factors within the same year and over short or long times up to a year. Many researches generally analyzed directly the changes in vegetation or land-use/cover over different periods; however, the influence of factors in different periods on vegetation or land use/cover are relatively rare [12,13,45]. In this study, the environmental factors affecting vegetation patterns at the three levels in 2007 were the same at different times when the vegetation map was published, indicating that shorter or longer time averages of environmental factors had similar effects on vegetation patterns. The reason might be that the three environmental factors slope, drainage class, and soil depth changed less during the last 20 years. However, the causal anthropogenic factors were different during the four periods, which may indicate a phenomenon for vegetation pattern changes in highly disturbed regions due to socio-economic development. Short-term factors had greater impacts on land use/cover and vegetation patterns than long-term factors in this study. These results were partly different from our hypothesis, i.e., short-term factors have greater impacts on land use/cover pattern than on vegetation pattern. The possible reasons might be the dominant vegetation and land use/cover are farmlands, which tend to be influenced by short-term factors.

### 4.2. Different Classification

Various plant communities are integrated into a certain hierarchical classification level according to their inherent natural characteristics, which makes the similarities in one type of vegetation and the differences in different types of vegetation significant [1]. The functional and structural properties of the types of vegetation at different classification levels are different, and these types of vegetation at different levels may respond differently to the same ecological variables. Compared with vegetation classification systems, land is divided into several different types according to the form and purpose of land use. Therefore, the land use/cover system is a system designed for the planning of land use and is more closely related to human activities.

Many vegetation classification levels, such as plant function types [46], Holdridge life zones [47], International Geosphere–Biosphere Program (IGBP) systems [48], and types of vegetation from the vegetation map of the People's Republic of China [10], were used to simulate vegetation distribution according to their environment. In this study, we also selected types of vegetation from the vegetation map of the People's Republic of China, i.e., vegetation groups reflecting similar appearances of communities, vegetation types reflecting similar appearances of communities and climates, and formations and subformations reflecting similar dominant species, as vegetation classification levels [1]. The explained variances in the levels of vegetation group, vegetation type, and formation and subformation were similar. The total variances explained by four similar variables, including slope, drainage class, soil depth, and gross domestic product per capita or gross domestic product from primary industry or Defense Meteorical Satellite Program/Operational Linescan System night light, reached 45% in the three vegetation classification levels in 1986–2007, 1991–2007, and 1996–2007 and even 53% in 2001–2007 (Tables 2–4). However, the total variances explained by the top four factors were lower than 22% in the land use/cover in the four periods. Temperature and precipitation factors played an important role in the land use/cover system but did not significantly affect the landscape pattern in most vegetation classification systems, indicating that different factors should be used in different classification systems when terrestrial carbon cycles are simulated and predicted under global warming.

*4.3. Other Possible Factors Not Considered in This Research*

Vegetation patterns are not only caused by geography, climate, soil, and human disturbance but are also influenced by other factors, such as crops planted in particular fields, duration of cultivation, government policy, and historical disturbance from humans, animals, and nature. However, complexity, quantitative limitations and the lack of historical information make it difficult to analyze the impact of these factors on vegetation [8]. Precipitation is a commonly used ecological factor in many studies, but in fact, altering rainfall time and interval with no change in total rainfall quantity led to changes in vegetation and other environmental factors [44]. Carbon dioxide fertilization, nitrogen deposition and even extreme weather events, including abnormal temperature and precipitation, can affect vegetation or land use patterns [12]. These climate factors may need to be considered in future research. Some soil characteristics, such as electrical conductivity, pH, very fine sand content, $CACO_3$, soil saturation, soil organic carbon, soil organic nitrogen, and C:N ratios, have been shown to be closely related to vegetation [17,27,49]. Among these soil factors, the same or similar factors, such as calcium carbonate, pH, and electrical conductivity, were adopted in this study, although they were not closely related to the vegetation pattern. However, this does not mean that they do not significantly affect vegetation, because these soil factors were removed due to their high correlation with other ecological factors (variance inflation factors greater than 10). Many ecological factors cannot be considered in isolation because of their strong interactions with one another [12].

In addition, both landscape metrics and vegetation–environment relationships are sensitive to the scale of sampling, i.e., the extent of observations in the study [17,50,51]. Scale-dependent relationships among many environmental variables, including biological and biophysical variables, have been demonstrated [28,30]. Understanding the scale-dependent relationship and considering scale is important when interpreting vegetation patterns. Finally, errors, such as algorithm errors of data preprocessing and survey errors of vegetation categories, are observed when generating vegetation categories and ecological factor data.

## 5. Conclusions

The causal factors for the land use/cover and vegetation classification systems were different, and the variance explained by causal factors was much higher in the vegetation classification system than the land use/cover system. Causal factors were different at

different times for the land use/cover system, and soil, temperature, and precipitation were the top three causal factors in 1986–2005, 1991–2005, and 2001–2005; in addition to precipitation and soil, the systems were also affected by anthropogenic disturbance and atmospheric factors in 1996–2005. For the three vegetation classifications, slope, drainage class, and soil depth were the top three impact factors in 1986–2007, 1991–2007, 1996–2007, and 2001–2007; in addition to the top three factors, anthropogenic disturbance was also a causal factor. Short-term factors had greater impacts on land use/cover and vegetation patterns than long-term factors. Different ecological factors should be considered in different classification systems and different levels in vegetation classification systems when natural resource management and urban or rural planning are conducted.

**Author Contributions:** Conceptualization, S.Y., X.L., B.L., J.G. and Y.Z.; methodology, S.Y. and Y.Z.; software, S.Y. and J.Z.; data curation, S.Y. and Y.Z.; writing—original draft preparation, S.Y. and Y.Z.; writing—review and editing, S.Y. and Y.Z.; visualization, S.Y. and Q.S.; project administration, L.L. All authors have read and agreed to the published version of the manuscript.

**Funding:** This work was funded by National Key R&D Program of China (2018YFC0506903).

**Institutional Review Board Statement:** Not applicable.

**Informed Consent Statement:** Not applicable.

**Data Availability Statement:** All data were available online. Land use/cover data for the study area in 2005 were obtained from the Resource and Environment Data Cloud Platform (http://www.resdc.cn/Default.aspx), and the vegetation distribution data in 2007 were obtained from the Vegetation Map of the People's Republic of China (Editorial Committee of Vegetation Map of China, the Chinese Academy of Science, 2007). Geographic variables were obtained from Zhao et al. (2018). The soil data were provided by the China soil map-based harmonized world soil database (HWSD). Population, gross domestic product, gross domestic product from primary industry, gross domestic product from secondary industry, and gross domestic product from tertiary industry in 1986–2007 were obtained from Nian.Jian.XiaZe.com (http://nianjian.xiaze.com/info/bjtjnj.html). The Defense Meteorological Satellite Program/Operational Linescan System night light data in 1992–2007 were obtained from the National Centers for Environmental Information (https://ngdc.noaa.gov/eog/download.html). The monthly average temperature and atmospheric variables in 1986–2007 were obtained from the National Meteorological Science Data Center (http://data.cma.cn/). The annual precipitation and mean annual temperature in 1986–2007 were provided by the Resource and Environment Data Cloud Platform (http://www.resdc.cn/Default.aspx). The maximum temperature, minimum temperature and precipitation for every month in 1986–2007 were provided by WorldClim (https://www.worldclim.org/).

**Acknowledgments:** We thank reviewers and the editor for their effort to review this manuscript.

**Conflicts of Interest:** The authors declare no conflict of interest.

## Appendix A

**Table A1.** Land use/cover and vegetation in three classification levels in the Jing-Jin-Ji region.

| Land Use/Cover | Vegetation Groups | Vegetation Types | Formations and Subformations |
|---|---|---|---|
| 1. Woodland | 1. Needleleaf forest | 1. Cold-temperate and temperate mountains needleleaf forest | 1. *Larix principis-rupprechtii* forest |
| | | 2. Temperate needleleaf forest | 2. *Pinus tabulaeformis* forest<br>3. *Platycladus orientalis* forest |
| | 2. Broadleaf forest | 3. Temperate broadleaf deciduous forest | 4. *Quercus mongolica* forest |
| | | | 5. *Quercus liaotungensis* forest<br>6. *Quercus aliena* forest<br>7. *Quercus acutissima forest*<br>8. *Quercus variabilis* forest |

Table A1. *Cont.*

| Land Use/Cover | Vegetation Groups | Vegetation Types | Formations and Subformations |
|---|---|---|---|
| 2. The grass | | | 9. *Robinia pseudoacacia* forest<br>10. *Salix matsudana* forest<br>11. *Populus simonii* forest<br>12. *Populus nigra* forest<br>13. *Populus davidiana* forest<br>14. *Chosenia arbutifolis, Populus suaveolens* forest<br>15. *Betula platyphylla* forest |
| | | 4. Temperate microphyllous deciduous woodland | 16. *Ulmus macrocarpa* woodland |
| | 3. Scrub | 5.Temperate broadleaf deciduous scrub | 17. *Corylus heterophylla* scrub<br>18. *Lespedeza bicolor* scrub<br>19. *Prunus armeniaca* var. *ansa* scrub<br>20. *Gleditsia heterophylla* scrub<br>21. *Vitex negundo* var. *heterophylla, Zizyphus jujuba* var. *spinosa* scrub<br>22. *Cotinus coggygria* var. *cinerea* scrub<br>23. *Spiraea spp.* scrub<br>24. *Ostryopsis davidiana* scrub<br>25. *Hippophae rhamnoides* scrub<br>26. *Rosa spp., Cotoneaster spp.* scrub<br>27. *Tamarix chinensis* scrub |
| | 4. Steppe | 6. Temperate grass-forb meadow steppe | 28. *Festuca ovina*, forb meadow steppe<br>29. *Stipa baicalensis*, forb meadow steppe<br>30. *Bothriochloa ischaemum*, forb meadow steppe<br>31. *Filifolium sibiricum*, grass-forb meadow steppe |
| | | 7. Temperate needlegrass arid steppe | 32. *Aneurolepidium chinense*, needlegrass steppe<br>33. *Stipa grandis* steppe<br>34. *Stipa krylovii* steppe<br>35. *Stipa bungiana* steppe<br>36. *Koeleria cristata, Agropyron cristatum*, dwarf needlegrass steppe<br>37. *Thymus mongolicus*, needlegrass steppe<br>38. *Artemisia frigida*, dwarf needlegrass steppe<br>39. *Artemisia gmelinii*, grass steppe<br>40. *Artemisia giraldii*, grass steppe |
| | 5. Grass-forb community | 8. Temperate grass-forb community | 41. *Bothriochloa ischaemum* community<br>42. *Themeda triandra* var. *japonica* community |
| | 6. Meadow | 9. Temperate grass and forb meadow | 43. *Arundinella hirta, Spodiopogon sibiricus*, forb meadow<br>44. *Imperata cylindrica* var. major meadow<br>45. *Carex spp.*, forb meadow |
| | | 10. Temperate grass, *Carex* and forb swamp meadow | 46. Contain *Carex spp.*<br>47. *Agrostis alba, Hordeum bogdanii* swamp meadow |
| | | 11. Temperate grass and forb holophytic meadow | 48. *Phragmites communis* holophytic meadow<br>*49. Achnatherum splendens* holophytic meadow<br>50. *Kalidinm spp., Puccinellia distans* holophytic meadow<br>51. *Suaeda glauca* holophytic meadow |

**Table A1.** *Cont.*

| Land Use/Cover | Vegetation Groups | Vegetation Types | Formations and Subformations |
|---|---|---|---|
| 3. Water area | 7. Swamp | 12. Cold-temperate and temperate swamp | 52. *Phragmites communis* swamp |
| 4. Cultivated land | 8. Cultural vegetation | 13. One crop annually and cold-resistant economic crops | 53. Spring wheat, middle and late crop soybean, corn, Chinese sorghum; sugar beet, sunflower, flux; apple (the seedling takes cover in winter) 54. Spring wheat, naked oats, buckwheat, potatoes; flux |
| | | 14. One crop annually, cold-resistant economic crops and deciduous orchards | 55. Spring (winter) wheat, Chinese sorghum, millet, gruel, *Medicago sativa*; sunflower, sugar beet; apple, pear, date, valnut 56. Winter wheat, corn, Chinese sorghum, millet, sweet potatoes; peanut; apple, pear, hawthorn, persimmon, walnut, chestnut, date, grape (takes cover in winter) |
| | | 15. Three crops two years and two crops annually non irrigation, deciduous orchards | 57. Winter wheat, corn, Chinese sorghum, sweet potatoes; cotton, tobacco, peanut, sesame; apple, pear, hauthorn, persimmon, walnut, pomegranate, grape |
| 5. The land of industry, mining and residence in urban and rural 6. Unused land | 9. No vegetation | 16. No vegetation | 58. No vegetation |

**Table A2.** Ecological variables.

| Categories of Variable | Variables |
|---|---|
| Geography | Elevation Aspect Slope |
| Soil | Texture class Available water storage capacity Drainage class Soil depth Subsoil base saturation Subsoil Calcium Carbonate Subsoil Gypsum The cation exchange capacity in subsoil Percentage clay in the subsoil The electrical conductivity of subsoil The exchangeable sodium percentage in subsoil Volume percentage gravel in the subsoil The percentage of organic carbon in subsoil pH in the subsoil Bulk density of subsoil Percentage sand in the subsoil Percentage silt in the subsoil Topsoil base saturation Topsoil Calcium Carbonate Topsoil Gypsum The cation exchange capacity in topsoil Percentage clay in the topsoil The electrical conductivity of topsoil The exchangeable sodium percentage in topsoil |

**Table A2.** *Cont.*

| Categories of Variable | Variables |
| --- | --- |
| Precipitation | Volume percentage gravel in the topsoil<br>The percentage of organic carbon in topsoil<br>pH in the topsoil<br>Bulk density of topsoil<br>Percentage sand in the topsoil<br>Percentage silt in the topsoil<br>Annual precipitation<br>Monthly precipitation in April and October<br>Precipitation of wettest month (i.e., Monthly precipitation in July)<br>Precipitation of driest month (i.e., Monthly precipitation in January)<br>Precipitation of warmest quarter<br>Precipitation of coldest quarter<br>Minimum temperature in April, July, and October<br>Maximum temperature in January, April, and October<br>Monthly average temperature in January, April, July, and October<br>Max temperature of warmest month (i.e., Maximum temperature in July)<br>Min temperature of coldest month (i.e., Minimum temperature in January) |
| Temperature | Mean annual temperature<br>Mean diurnal range<br>Isothermality<br>Temperature annual range<br>Mean temperature of warmest quarter<br>Mean temperature of coldest quarter |
| Atmospherics | Water vapor pressure in January, April, July, and October<br>Wind speed in January, April, July, and October<br>Population<br>Gross domestic product |
| Anthropogenic disturbance | Gross domestic product from primary industry<br>Gross domestic product from secondary industry<br>Gross domestic product from tertiary industry<br>Gross domestic product per capita<br>Gross domestic product per unit area<br>Defense meteorological satellite program/operational linescan system night light |

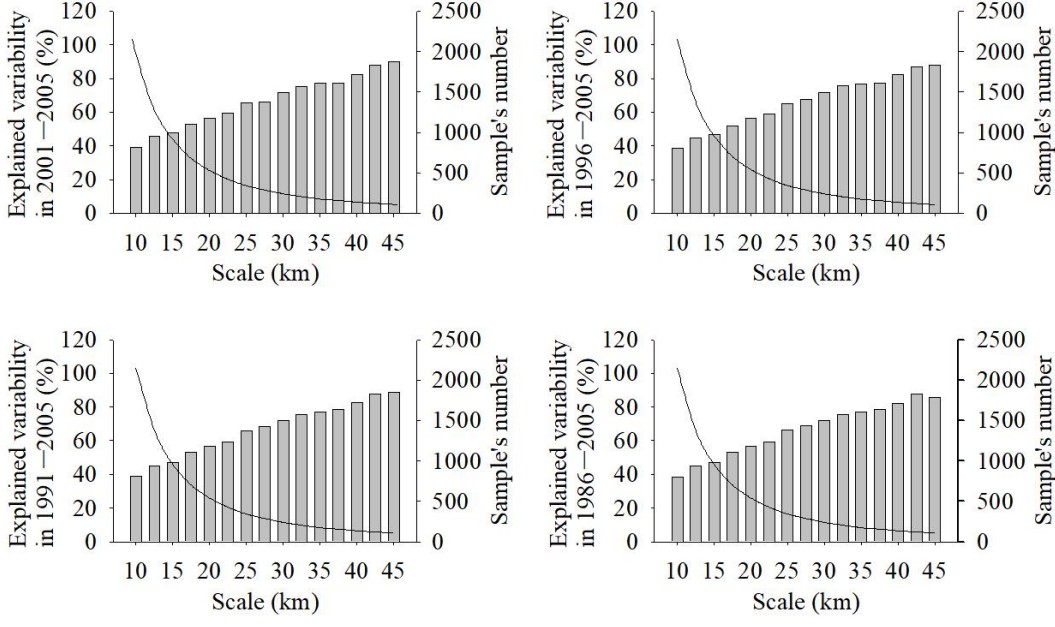

**Figure A1.** The number of samples and explained variance at all scales in land use/cover in four periods.

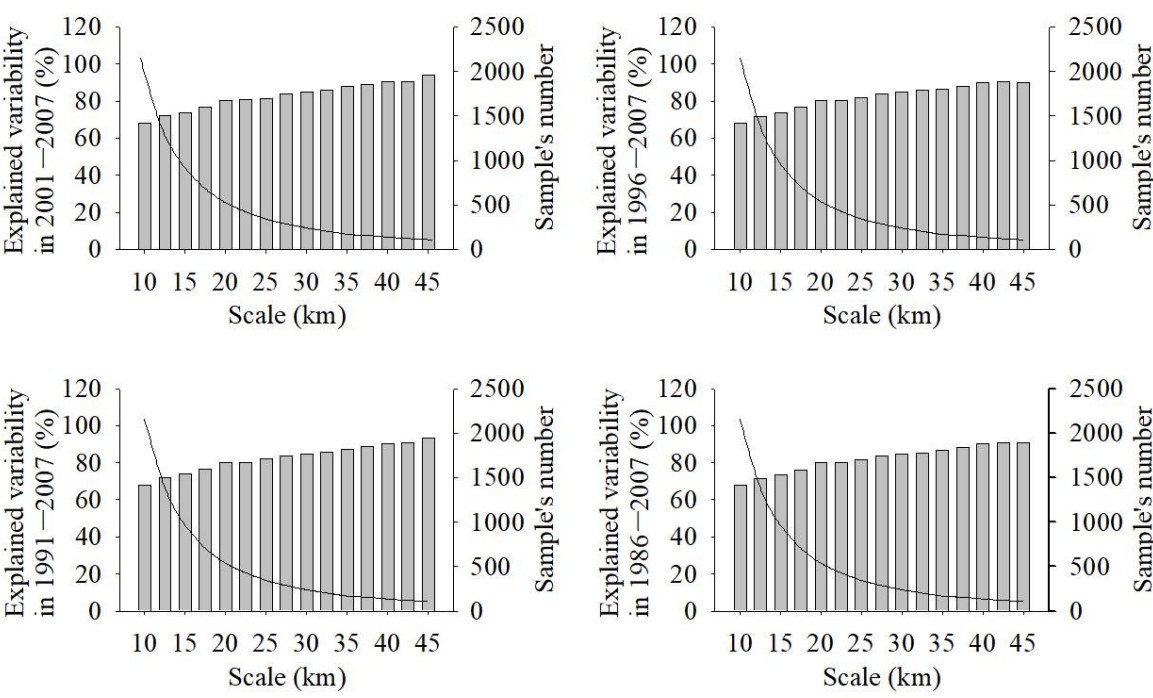

**Figure A2.** The number of samples and explained variance at all scales in vegetation groups in four periods.

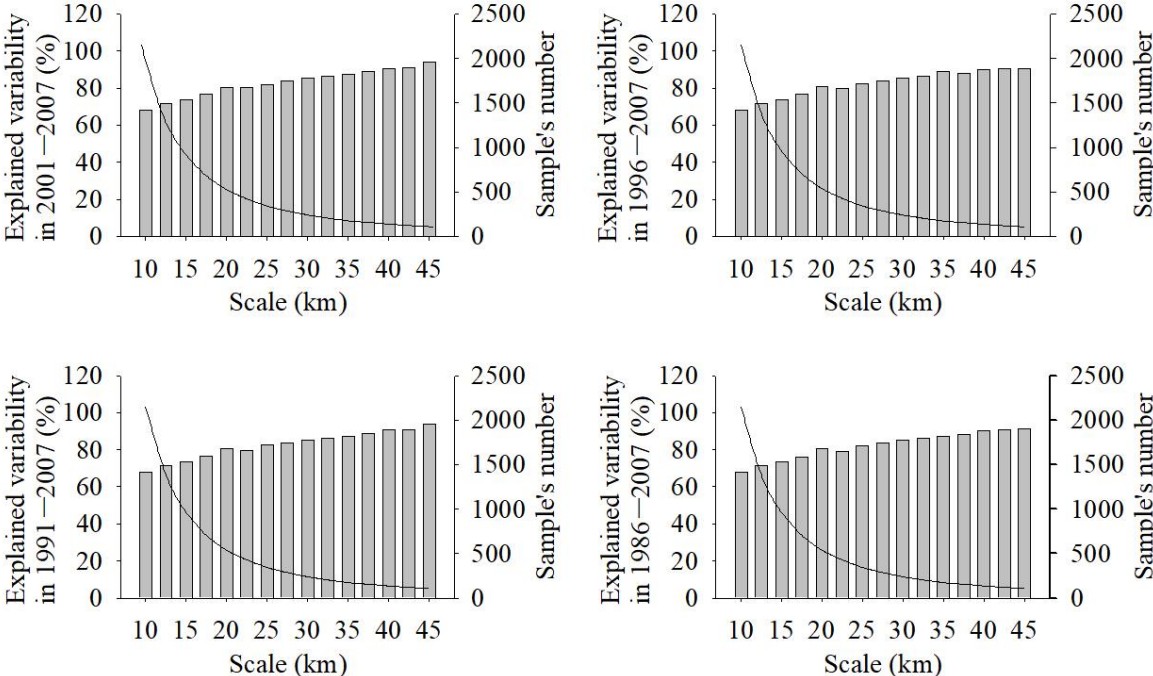

**Figure A3.** The number of samples and explained variance at all scales in vegetation types in four periods.

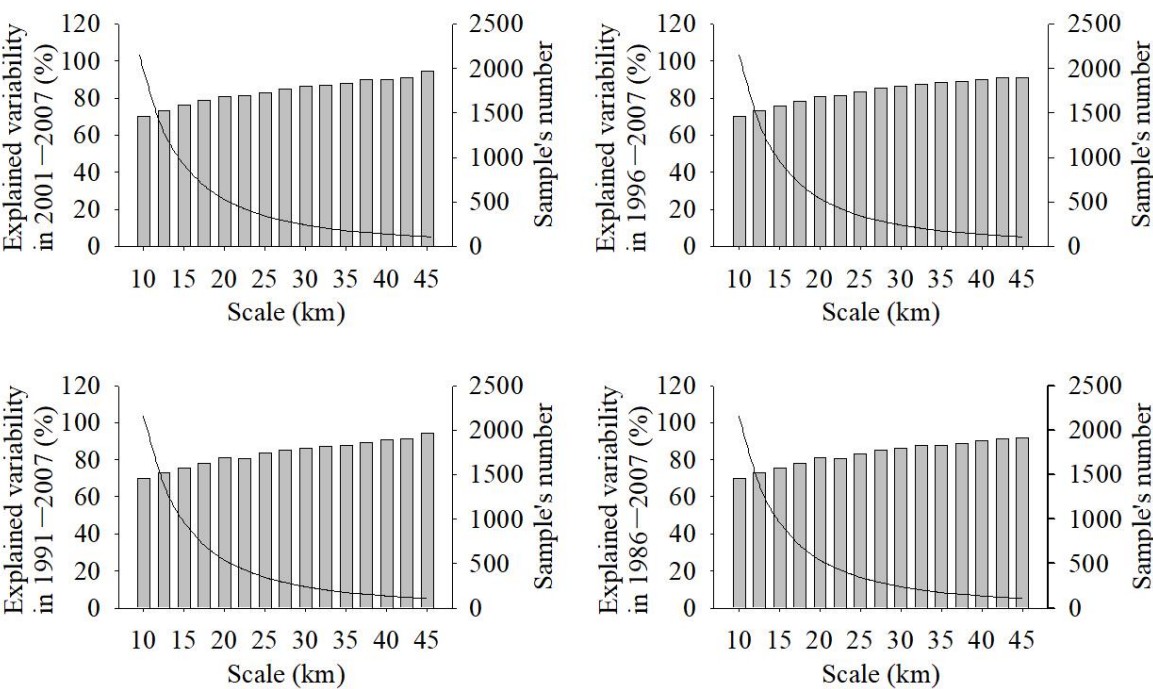

**Figure A4.** The number of samples and explained variance at all scales in formations and subformations in four periods.

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
