# Peer review of "Different Causal Factors Occur between Land Use/Cover and Vegetation Classification Systems but Not between Vegetation Classification Levels in the Highly Disturbed Jing-Jin-Ji Region of China"

_sustainability, doi:10.3390/su13084201_

Round 1
Reviewer 1 Report
This study is interesting attempt to find relationship between vegetation types, land use/cover and both ecological and socioeconomic factors. I am not sure whether statistical analyses were run properly. Therefore I suggest revising this manuscript.
Specific commnents:
Abstract:
The sentence “Different classification systems should be used in global warming and carbon cycle simulations depending on the specific purpose” is not clear.
Replace “impact factor” with another word
Introduction:
“....including various types of natural covers (such as soil, vegetation and wetlands) and artificial covers (such as buildings, roads and reservoirs)”. – wetland can be also type of vegetation. Reservoir – what do you mean lakes are natural but fish pond is artificial.
The paragraph starting from the sentence “Methods used to reveal the relationship generally include ordination techniques and related statistical models” is about ordination analyses. Authors should add that these statistical tools are used at much smaller level – level of patch and examine relationship between species composition and local environmental factors. The paragraph before is about NVDI that is used at higher scale.
Authors wrote that there are constrained and unconstrained ordinations. However, there are also eignevalue-based and distance-based ordination. PCoA and NMDS were not mentioned.
“and the results show that vegetation is influenced by many environmental factors, including soil, lithology, geography, temperature, precipitation, wind and human historical disturbance” – light, especially in forest, is very important.
The hypothesis: “We hypothesized that the main causal factors for the distribution pattern of land use/cover are socioeconomic factors while those for vegetation are environmental factors” sounds very obvious. I miss hypothesis connected with changes in time.
Material and methods & Results
Authors claimed that factors are not correlated in RDA. It is hard to believe it looking at figures 3-5. It is not clear how many samples were subjected to RDAs. Authors wrote that “The number of samples ranged from 107 to 2150 at all scales in the land use/cover system and 3 levels of the vegetation classification system”. How many samples were used as “species variables”. Perhaps model building using AIC would be better option because if there are many samples almost all applied ecological factors are statistically significant.
“Landscape metrics served as species variables, and ecological factors served as environmental variables”. I guess not only landscape metrics but vegetation data as well.
In ordination analyses that using cover-abundance data of species, species variables represent the same variable. In this case it is not certain what type of data was used (categorical, integer, continuous, what are ranges?). It can affect results. I suggest scaling all variables both those used as “species variables” and all environmental factors.
Table 1: F-value it is pseudo-F because permutation test was used.
Figures- 2-5 show only inter-correlations among used landscape metrics, vegetation metrics and environmental factors. Show as it is possible samples as points. Different colors could be used for landscape and vegetation metrics and environmental factors. Type of line is not clear visible.
Discussion is long and does not concentrate enough on testing hypothesis and aims of the study.
Reviewer 2 Report
The study analyses a conspicuous set of environmental and human-related variables as possible causal factors of land use and vegetation diversity in a region of northern China. The work is well-conducted, in line with the scope of the journal, and interesting for the scientific audience. The only major issue to be addressed regards the style of the English language, that needs a considerable improvement. For the sake of understandability, I strongly recommend a revision by a native speaker or at least by an expert. For the rest, I have some minor comments that I detailed below.
Overall, I consider the manuscript acceptable for publication after minor revisions.
Citations and references: Throughout the manuscript, citations are not given correctly according to the style of the journal (only numbers in order of first appearance should be reported in the text), and the reference list should be ordered respectively.
Abstract and elsewhere throughout the manuscript: “atmosphere” and “atmospheric factors” – aren’t they part of climatic factors? I think that the fact that you split them apart needs more justification. I also suggest to use always the expression “atmospheric factors” instead of “atmosphere”.
Keywords: Do not repeat here words that are already in the title.
Introduction
“…such in the Atlantic Ocean…” I guess you mean in countries on the Atlantic Ocean?
“…the results show that vegetation is influenced by many environmental factors…” here it is appropriate to cite also a couple of examples of studies on synanthropic vegetation. I suggest:
- Nowak A, Nowak S, Nobis M, Nobis A (2015) Crop type and altitude are the main drivers of species composition of arable weed vegetation in Tajikistan. Weed Research 55(5): 525–536. https://doi.org/10.1111/wre.12165
- Fanfarillo E, Petit S, Dessaint F, Rosati L, Abbate G (2020) Species composition, richness, and diversity of weed communities of winter arable land in relation to geo-environmental factors: A gradient analysis in mainland Italy. Botany 98(7): 381–392. https://doi.org/10.1139/cjb-2019-0178
“…to better manage and plan natural resources…” also considering that the Chinese Government led major afforestation projects in the region in the last years. See e.g. Jin et al., Forests 12(3): 316, 2021.
Data and method
Study area: “…and mean annual temperature ranging from -3°C to 14°C…”.
“…water vapor pressure and wind speed variables…” and later in 2.4 “…including precipitation, temperature, atmosphere…” see my comment above.
“Landscape metrics served as species variables…” as response variables? “Species variables” is really not appropriate here.
Table A1: move the classes in the first column in correspondence of the respective ones in the other columns.
Round 2
Reviewer 1 Report
The authors considered all my comments and made appropriate changes or when they disagreed then provided rationale.
I've got only minor comments.
The hypothesis "We hypothesized that short-term factors have great impacts on land use/cover pattern and long-term factors have great impacts on vegetation pattern" still sound strange and does not depict problem. I recommend to change e.g. "short-term factors have greater impacts on land use/cover pattern than..." or otherwise. Further, this has to be addressed in the discussion chapter.
